# Alternative Targets for sPLA2 Activity: Role of Membrane-Enzyme Interactions

**DOI:** 10.3390/membranes13070618

**Published:** 2023-06-23

**Authors:** Anna S. Alekseeva, Ivan A. Boldyrev

**Affiliations:** Shemyakin-Ovchinnikov Institute of Bioorganic Chemistry, Russian Academy of Sciences, 117997 Moscow, Russia; anna@lipids.ibch.ru

**Keywords:** phospholipase A2, PLA2, lipid membrane, enzyme inhibitors, interfacial enzymes

## Abstract

The secreted phospholipases A2 (sPLA2s) play important roles both physiologically and pathologically, with their expression increasing significantly in diseases such as sepsis, inflammation, different cancers, glaucoma, obesity, Alzheimer’s disease and even COVID-19. The fact has led to a large-scale search for inhibitors of these enzymes. In total, several dozen promising molecules have been proposed, but not a single one has successfully passed clinical trials. The failures in clinical studies motivated in-depth fundamental studies of PLA2s. Here we review alternative ways to control sPLA2 activity, outside its catalytic site. The concept can be realized by preventing sPLA2 from attaching to the membrane surface; by binding to an external protein which blocks sPLA2 hydrolytic activity; by preventing sPLA2 from orienting properly on the membrane surface; and by preventing substrate binding to the enzyme, keeping the catalytic site unaltered. Evidence in the literature is summarized in the review with the aim to serve as a starting point for new types of sPLA2 inhibitors.

## 1. Introduction

Secreted phospholipases A2 (sPLA2s) represent a large lipolytic family of interfacial enzymes with different physiological functions and tissue distribution patterns, unified by their ability to catalyze the hydrolysis of the ester bond at the sn-2 position of phospholipids in membranes. Upon hydrolysis, sPLA2s release free fatty acids and generate lysophospholipids, while reaction products can be converted into a wide variety of bioactive molecules [1]. The sPLA2s play important roles both physiologically and pathologically, with their expression increasing significantly in diseases such as sepsis, inflammation, different cancers, glaucoma, obesity, Alzheimer’s disease [2] and even COVID-19 [3]. Studies using transgenic and knockout mice for several sPLA2 enzymes, in combination with lipidomic approaches, helped to highlight their distinct contributions to various biological events [4]. The existence of inactive zymogens of sPLA2 in the pancreas [5] is one of the ways to regulate enzyme activity in mammals, and the regulation of mRNA levels is another [6].

The involvement of sPLA2s in the development of various pathologies has led to a large-scale search for inhibitors of this type of enzyme. In total, several dozen promising molecules have been proposed (see [7,8,9]), but not a single one has successfully passed clinical trials [9]. All proposed molecules bind to the catalytic site of the enzyme. The cases of darapladib and varespladib are illustrative for small-molecule PLA2 inhibitors.

Darapladib is a lipoprotein-associated phospholipase A2 (LpPLA2) inhibitor that was developed to treat atherosclerosis [10]. Clinical studies were conducted by GlaxoSmithKline. In a study of 16,000 patients with acute coronary syndrome (ACS), darapladib did not reduce the risk of death from coronary heart disease, myocardial infarction and urgent coronary revascularization compared with a placebo in patients with ACS receiving standard care. This failure indicates that blocking catalytic activity is not sufficient to control LpPLA2, or that the role of LpPLA2 in the development of inflammation is not limited to lipid hydrolysis.

Varespladib and methylvarespladib are inhibitors of secretory phospholipase A2. The start of the clinical trials was positive. Clinical studies in patients with coronary artery disease showed significant reductions in levels of sPLA2 and cholesterol (in low-density lipoprotein), as well as C-reactive protein (CRP) and other biomarkers of inflammation [11,12,13]. A phase III study was launched in 2010 to evaluate the safety and efficacy of short-term treatment with methylvarespladib in patients with ACS. The study was stopped in March 2012 due to futility and possible harm [14]. Nevertheless, varespladib continued on its way to the clinic, but with a change in nosology. Now it is considered as an antidote for snake bites [15,16]. The latter fact demonstrates the unique feature of secretory PLA2—a large homology of enzymes from different organisms.

The failures in clinical studies of classical inhibitors indicates that our lack of knowledge of PLA2s remains high. In fact, recent reviews on the topic state that the development of new selective inhibitors requires a deeper understanding of PLA2s, and this requires in-depth fundamental studies of these enzymes [9,17]. The latter are currently being undertaken. Regions outside the catalytic site of sPLA2 suitable for inhibition of enzyme activity are the focus of the present review. Some of them are newly discovered, while others are closer to application. For more information on classical small-molecule inhibitors of the sPLA2 catalytic site, readers are referred to the excellent reviews [7,9,17]. Here we review alternative ways to control sPLA2 activity and discuss concepts and possibilities for future sPLA2 inhibitor development.

## 2. Directions to Attack sPLA2 to Control its Enzymatic Activity

The sPLA2s are small water-soluble enzymes, around 15 kDa in mass. To reach a lipid substrate, sPLA2 should attach to the lipid membrane, properly orient on the membrane surface and bind the substrate (pull it from the bilayer). These steps are schematically shown in Figure 1. Once the substrate is bound to the catalytic site, hydrolysis occurs.

Because sPLA2 works only at the water–lipid interface, acting on phospholipid assemblies rather than on single phospholipids, the enzyme has two binding sites. One is the catalytic site responsible for substrate hydrolysis. The other one is the interface-binding site (IBS) used to connect the enzyme to the bilayer. The sPLA2 catalytic site is highly conserved, while the membrane-binding site, on the contrary, differs from enzyme to enzyme [18].

Significant efforts were put toward the development of classic small-molecule inhibitors, which bind to the catalytic site, making hydrolysis impossible. According to Figure 1, there are other possibilities to prevent hydrolysis. The first one is to prevent sPLA2 from attaching to the membrane surface (Figure 1, process 1). The second one is to prevent sPLA2 from orienting properly on the membrane surface (Figure 1, process 2). In this case the catalytic site remains out of contact with the lipid bilayer, and thus the substrate cannot be bound. The possibility is based on the recent finding of multiple membrane-binding sites on the surface of sPLA2, only one of which is associated with the hydrolysis process [19]. The third possibility to prevent hydrolysis is to prevent a substrate from binding to the enzyme, keeping the catalytic site unaltered (Figure 1, process 3). The idea is based on the fact that the substrate-binding pocket is bigger than the catalytic site itself. Thus, altering, for example, entrance region one could prevent substrate binding and subsequent hydrolysis.

sPLA2-driven hydrolysis is described by the classical scheme of interfacial kinetics (Figure 2). According to the scheme, an sPLA2 could be inhibited either by water-soluble inhibitors (I_s_) in the aqueous phase or at the membrane interface by membrane-bound inhibitors (I_i_).

Below we discuss the possibilities to control sPLA2-driven lipid hydrolysis through acting on processes 1, 2 and 3 (Figure 1), but not hydrolysis itself (process 4).

## 3. Preventing sPLA2 Binding to the Target Membrane

Because the membrane-binding step is required for the hydrolysis of sPLA2, it is possible to prevent hydrolysis simply by preventing the enzyme from binding to the target bilayer. That is to prevent step 1 (Figure 1) or, in classical notation, to inhibit the E-to-E* process (Figure 2). For this purpose, membrane mimetics (micelles, lipid nanoparticles and liposomes) with similar structures of water–lipid interface could be used. Thus, to prevent sPLA2 from binding to the bilayer, an unhydrolysable membrane is added to the system. This holds the enzyme bound, keeping it away from the target membrane.

The phenomenon was described at the rise of the study of the biochemistry of sPLA2. In the 1990s it was used to study the subtleties of the kinetic parameters of sPLA2 hydrolysis. M. Gelb and co-authors had proposed the usage of an “imperfect neutral diluent” which is added to the hydrolysable lipid matrix at a mole fraction of 0.1–0.3 (a higher mole fraction may disrupt the vesicles). An imperfect neutral diluent is a lipid-like substance that is not hydrolysed by the enzyme [20,21]. A neutral diluent is an amphiphile that forms aggregates to which PLA2 binds, but the amphiphile does not occupy the active site of the bound enzyme. Thus, neutral diluents, when incorporated into phospholipid vesicles, were useful as two-dimensional solvents for decreasing the mole fraction of substrate at the interface.

To understand the principles of sPLA2 interaction with unhydrolysable membranes, we should go deeper into the pioneering works by Prof. Gelb and co-authors. In these works, DMPM (1,2-dimyristoyl-sn-glycero-3-phosphomethanol), a synthetic lipid with a negative charge on the polar part, most often acts as a substrate (a hydrolysable lipid). The course of the reaction was typically monitored by the pH-stat method [22].

Neutral diluents for sPLA2 can be identified by monitoring the rate of alkylation of the active site histidine residue by phenacyl bromides [21]. If the enzyme binds to the aggregate of the neutral diluent and if the rate of active-site alkylation of the bound enzyme is the same as for the enzyme in the aqueous phase, then the active site is not occupied by a molecule of the aggregate. Ligands that bind to the active site of an enzyme bound to a neutral diluent confer protection from alkylation, and this effect was used to measure the equilibrium constant for the dissociation of the enzyme–ligand complex [20]. For this purpose, liposomes with different mole fractions of neutral diluent were tested, and the rate of hydrolysis was monitored.

For each sPLA2, neutral diluents should be selected individually. For example, the amphiphiles 1-palmitoyl-2-deoxy-sn-glycero-3-phosphocholine and rac-2-hexadecyl-sn-glycero-3-phosphocholine that are effective neutral diluents of the sPLA2 from the porcine pancreas [21] and hexadecylphosphorylcholine which is a neutral diluent for the sPLA2 from human synovial fluid [23] are not neutral diluents for bee venom PLA2 (bvPLA2), nor is bis-PC. Additional complexity exists regarding bvPLA2 because the binding of ligands to the active site does not fully protect the enzyme from alkylation; the bvPLA2–ligand complex is only about 80% protected from alkylation, which is in marked contrast to pancreatic and synovial fluid sPLA2s, which are nearly fully protected in complexes with active-site ligands. At the same time, liposomes composed from ditetradecylphosphomethanol (DTPM) work as a trap for bvPLA2—the enzyme is adsorbed onto the surface of the particles, and the addition of DMPM to the system does not lead to the onset of hydrolysis (i.e., the bvPLA2 enzyme remained bound to the particles consisting of DTPM). When using DTPC (neutral charge), bvPLA2 binding was too low to prevent hydrolysis [20].

Through such studies, it has been demonstrated that the effects of many previously reported PLA2 inhibitors are due to a shift in the E-to-E* equilibrium (Figure 2) toward the enzyme in the aqueous phase [21,24]. The major problems encountered in the assay of PLA2 activity have to do with the inability to control the E-to-E* equilibrium.

When describing the kinetics of sPLA2 hydrolysis, it is common to distinguish two extreme variants of enzyme behaviour: the hopping mode (i.e., the enzyme is desorbed from the membrane surface after each round or several rounds of hydrolysis) and the scooting mode (when the enzyme performs many rounds of catalysis after sorption onto the membrane). One or another mode of hydrolysis is realized depending on the concentrations of the enzyme and substrate (their ratio), the constants of desorption and other factors [25]. The residence time of the enzyme at the interface is not constant over the time course as the reaction progresses.

The ability to fix the enzyme on the particle surface can only be realized in the case of the scooting mode. The enzyme should hop from (or does not hop to) a hydrolysable membrane and should scoot over a nonhydrolysable membrane. Thus, it is removed from the reaction. This is the reason for the use of liposomes (sufficiently large, not micelles and not too small) with a negative charge (DMPM or similar), as for most sPLA2s the affinity for the negative charge of the membrane is greater than for zwitterions.

Today, the classic kinetic models developed by Gelb and co-authors are in service again. It has recently been shown that subcutaneous administration of lysolipids to rats led to a decrease in the activity of sPLA2 in the blood [26]. In addition, the inhibitory effect of detergent micelles on lipoprotein-associated PLA2 has been shown [27]. In both cases we assume that micelles formed by lysolipids or detergents are responsible for PLA2 binding and subsequent enzyme take-off from the reaction. The works by Cunningham et al. and Zhuo et al. show that blocking PLA2 binding to the membrane surface is a promising approach for PLA2 activity regulation in vivo.

## 4. sPLA2-Binding Proteins with Inhibitory Properties

The interface-binding site is not the only site on the sPLA2 surface which can put the enzyme into interaction with other molecules [28]. Since the 1980s, evidence of the interaction of sPLA2 with various molecules (not only with the lipid substrate) has appeared, and the number of sPLA2-binding molecules continues to grow.

sPLA2-binding proteins can act in a way similar to nonhydrolysable membranes—when bound to the IBS of the sPLA2, they prevent the enzyme from interacting with the target bilayer. Alternatively, they can translocate sPLA2, thus moving it away from the membrane. The latter process could formally be described as inhibition in solution (E ⇄ E·Is, see Figure 2).

Different parts of sPLA2 can act as binding sites for proteins—these are the IBS, N-terminal and C-terminal regions and β-loop. Binding to sPLA2 has a physiological significance (role or consequence)—it is the inhibition of enzyme activity or its activation, enzyme lysis or translocation in the cell and tissues [29]. Being associated with specific proteins, sPLA2 may not exhibit hydrolytic activity.

Several structural features allow sPLA2s to bind to structurally very diverse targets. Despite the generally compact structure of sPLA2s, the flexibility of the exposed side chains of the amino acids at the IBS promotes their optimal binding to phospholipid aggregates [18,30]. The flexibility of the sPLA2 β-loop might be another feature that favours their high adaptability to different proteins. For example, the β-loop of sPLA2s should be involved in the interactions of sPLA2s with β and γ PLA2-inhibitory proteins (β- and γ-PLIs) [31].

sPLA2-binding proteins can be combined into groups according to separate repetitive motifs (but this does not always guarantee binding), but it is rather conditional, as sPLA2-binding proteins are very diverse. For example, one of the characteristic motifs is a CRD (C-type carbohydrate recognition domain), which is part of the M-type sPLA2 receptor (and the CRD determines binding to it [32]).

The M-type sPLA2 receptor is a membrane glycoprotein with a very large extracellular domain and a single transmembrane domain [33]. A soluble form of the receptor was also found in mice and humans [34,35]. For the set of mouse sPLA2s, the properties of binding to different types of mouse M-type PLA2 receptors were investigated [36]. To assess binding affinity, researchers used a competition binding assay with [^125^I]-labelled OS 1 (*Oxyuranus scutellatus scutellatus* toxin 1, an sPLA2 with a proven high affinity to the, M-type receptor). Seven of the eleven tested sPLA2s showed high or moderate affinity and three were unable to bind to the different mouse receptors. There were no significant differences in binding affinities between the soluble and membrane-bound forms [36]. When bound to the soluble M-type sPLA2 receptor, sPLA2 becomes inactive [36,37].

When sPLA2s are bound to proteins, such complexes have been suggested to be involved in inflammation [38,39], hormone release [40,41] and neurotoxicity [42,43]. It has already been confirmed that sPLA2s are involved, through interaction with other proteins, in cell proliferation [44], antibacterial activity [45] and blood coagulation [46]. That is the reason the design of protein inhibitors should consider possible negative effects of sPLA2–protein complexes. Careful investigation of such complexes is required.

The M-type sPLA2 receptor has been shown to regulate the activity of sPLA2s in two ways: through acting as an sPLA2 inhibitor (in its soluble form); and through mediating sPLA2 endocytosis (in its membrane-bound form). After binding to the M-type PLA2 receptor, the complex is internalized by the cell through a clathrin-dependent pathway, and sPLA2 can then be degraded in lysosomes [47]. Endosomal vesicles could serve as a vehicle for delivering sPLA2 to specific intracellular compartments, where sPLA2, before being degraded in lysosomes, could manifest its enzymatic activity. Another possibility is that the internalization of the M-type sPLA2 receptor could serve a clearance function, selectively withdrawing sPLA2 from the extracellular fluid [48]. Pulmonary surfactant protein A (SP-A) belongs to the CRD-containing family of proteins, and it inhibits the enzymatic activity of GIIA and GX sPLA2s in pulmonary surfactant [49]. We suggest that in this case, inhibition at the interface takes place (E* ⇄ E·Ii, Figure 2). In any case, it is not clear whether the IBS is hidden or the enzyme is improperly oriented on the membrane surface (process 2 in Figure 1).

Soluble proteins that inhibit sPLA2s have been isolated from the blood of snakes and some other animals. These are ecologically connected with venomous snakes and are known as PLA2-inhibitory proteins, or PLIs [50]. The PLIs that interact with sPLA2s and inhibit their enzymatic activity have been purified from the sera of venomous (Crotalinae, Viperidae and Elapidae) as well as non-venomous snakes (for details, see recent reviews by Sanchez et al. and Santos-Filho et al. [51,52]). In this case, inhibition of sPLA2 in solution takes place (E ⇄ E·Is, see Figure 2). Three types of PLIs have been classified: C-type lectin-like proteins (α-PLIs), molecules bearing leucine-rich repeats similar to human α2-glycoprotein (β-PLIs) and those with a three-finger configuration (γ-PLIs) of acidic glycoproteins. Proteins responsible for resistance of snakes to their own venom were found in the blood of these animals; these proteins do not belong to the Ig family. The snake venom-poisoning resistance of some animals feeding on snakes, such as the opossum, mongoose and hedgehog, has been linked to similar proteins in their blood.

The sPLA2-binding element in α-PLIs probably resides in the C-type CRD-like domain, the most conserved portion of α-PLI molecules, comprising 60–70% of the sequence of each subunit. Type-α PLIs bind specifically to group II, but not group I or III, sPLA2s. The β-PLI sequence is unique, showing no significant homology to the α-PLIs; it contains 9 leucine-rich repeats (LRRs), each of 24 amino acid residues, encompassing over two-thirds of the molecule [53]. γ-PLIs are distributed more widely than α- and β-PLIs; they display broader sPLA2 specificity, being able to bind to group I, groups II and also group III sPLA2s.

The mechanism of action of PLIs has not been elucidated, and there have been attempts to create synthetic analogues based on PLIs, especially peptides [50].

We can summarize here that sPLA2 interactions with different proteins are well studied, and the data on PLA2–protein interactions could serve at least as a starting point in the search for peptide or protein agents capable of modulating sPLA2 activity.

## 5. Blocking the Enzyme in the Wrong Orientation on the Membrane Surface

Recently, Kuzmina and co-authors [19] had reported that the bvPLA2 enzyme has three different membrane-binding modes (Figure 3). This finding was obtained independently in “wet” experiments using atomic force microscopy and in silico experiments using molecular dynamics simulations. Switching between modes is responsible for the enzyme moving along the membrane, substrate uptake and product release to the biological milieu.

Figure 3 shows different orientations of PLA2 on the bilayer surface. Only mode 1 is responsible for the substrate uptake and subsequent hydrolysis. Because the enzyme appears to switch between modes and because there is only one mode which is responsible for substrate binding, the fixation of the enzyme in mode 2 or mode 3 could prevent hydrolysis. This is schematically depicted in Figure 1 as process 3 and could also be treated as inhibition at the interface (E* ⇄ E·Ii, Figure 2).

To clear up the possibility to fix the enzyme in mode 2 or mode 3, we should first discuss the role of sPLA2’s Trp128 residue in enzyme binding to the bilayer. The tryptophan residue is important in the formation of enzyme–bilayer contact. In general, tryptophan plays an essential role in the interfacial binding and activity of different sPLA2s [54,55]. Typically, sPLA2s that contain tryptophan in the IBS display the highest activity toward neutral phospholipid substrates [56,57], and the addition of tryptophan to the IBS significantly enhances the overall enzymatic activity [58].

In their recent work, Nasri et al. investigated the role of Trp128 in bvPLA2 activity [59]. The authors converted Trp128 to N’-formylkynurenine by the action of singlet oxygen (with the remaining protein staying intact). The modification reduced lipid hydrolysis by 80%. In the case of bvPLA2, Trp128 is not involved in the IBS (mode 1) [30] but is involved in mode 2 (see also Figure 4) [19]. That is the reason that protein binding to the bilayer surface could be traced by tryptophan fluorescence [20,30,60,61] (upon binding to the bilayer, the tryptophan residue faces a change in the polarity of the environment, with a corresponding change in the fluorescence). It is then assumed [19] that formylkynurenine, being a reactive species, reacts with an amino group of the DOPE lipid (involved in the experiments by Nasri et al.), thus stabilizing mode 2 and depopulating mode 1, preventing hydrolysis. In the experiment of Nasri et al., the enzyme was fixed in mode 2 using a chemical reaction between oxidized Trp128 residue and the DOPE lipid. This proves the concept that one can prevent hydrolysis by preventing the enzyme from adopting the proper orientation on the bilayer surface. At the same time, chemical modification of the enzyme with subsequent reaction with a lipid is not perfect solution to controlling the enzyme activity. One should find a more general approach, for example, by designing a specific peptide (lipopeptide) which could bind to the enzyme and fix it in mode 2.

On the other hand, membrane composition could affect the enzyme distribution between modes. In the light of the works by Nasri et al. [59] and Kuzmina et al. [19], a change in the Trp128 fluorescence in bvPLA2 could be assigned to the change in protein distribution between modes. Thus, what is assigned to optimal binding (e.g., in [62]) is probably the biggest fraction of mode 2. In mode 3, the enzyme contacts the bilayer through the β-loop (Figure 3 and Figure 4). The role of the β-loop of bvPLA2 in supporting interfacial catalysis was examined with the D99–118 deletion mutant by Ghomashchi et al. [63]. This mutant refolded with about half the yield obtained for the WT and displayed kinetic and vesicle-binding properties virtually identical to those of the wild type. All secreted PLA2s have this β-loop, but its function has not been identified [63] (D99–118 deletion is depicted in Figure 4). In the case of bvPLA2, the β-loop does not play a significant role in interfacial binding and catalysis on anionic interfaces, but it was assumed that it may play a role in product release to the aqueous milieu [19]. Mode 3 has the smallest contact area with the membrane (and corresponding energy of interaction), and the desorption of the enzyme from the membrane is the most probable. Thus, switching to mode 3 may be important for activating hopping of the enzyme from bilayer to bilayer. Among the amino acids involved in the formation of contact with the bilayer in the mode 3, Arg108 has the highest contact intensity (see Figure 4 to locate Arg108). It is important that the area around Arg108 (including Arg82) is responsible for binding with integrin ανβ3 [44] (here, residues’ numeration is presented according to the generalized approach of Renetseder et al. [64]; Saegusa et al. assign these as Arg100 and Arg74, respectively [44]). Ye at al. have designed peptide inhibitors (Cmpd8 and Cmpd21) which bind to Arg108 and Arg82 and prevent PLA2 from interacting with integrin ανβ3 [65]. At the same time, the catalytic activity of the PLA2 was not altered.

Simple binding of the enzyme in mode 3 does not terminate hydrolysis. It is necessary that the binding leads to fixing the orientation on the surface of the bilayer. In the case of the Cmpd8 and Cmpd21 peptides, they should be grafted onto the membrane surface (for example, by derivatization with hydrophobic moieties).

## 6. Blocking and Modulating Substrate Binding

The sPLA2 catalytic site is highly conserved. sPLA2s from different species have almost the same catalytic site. At the same time, the area surrounding the catalytic site is not so conserved. Thus, targeting inhibitors at the area surrounding the catalytic site could improve the specificity of inhibitor candidates. The cases below describe mainly inhibition in solution (E ⇄ E·Is, see Figure 2).

An inhibitor could either bind to the enzyme in regions around the catalytic site—thus preventing the main lipid substrate from binding due to the lack of ‘space’—or bind to cavities responsible for lipid tail selection. In the latter case, an inhibitor should modulate the enzyme’s selectivity towards lipid tail structures. For example, it could decrease the hydrolysis rate of arachidonoyl lipids while keeping the hydrolysis rate of other lipids unaltered.

Several areas on the PLA2 surface could be considered for this purpose. Recently, Mouchlis et.al. [66] have discovered unique hydrophobic cavities in sPLA2 and iPLA2 enzymes which are responsible for acyl chain specificity. The authors have used extensive molecular dynamics simulations of different PLA2s acting on multiple substrates. During simulations, several aliphatic residues including Val12, Leu102, Leu98 and Lue94 appeared to be involved in binding to a myristic chain (e.g., in palmitoyl-myristoyl-phosphatidylcholine, PMPC). These residues showed lower flexibility during the simulation with PMPC because they are part of the hydrophobic pocket to which the myristic acid is bound. At the same time, aromatic residues like Tyr21, Tyr24 and Tyr105 exhibited lower flexibility because they interact with the double bonds of arachidonic and linoleic acid (e.g., in palmitoyl-arachidonoyl-phosphatidylcholine, PAPC).

Interesting facts regarding inhibition have recently been discovered for LpPLA2, a member of another subfamily of PLA2. A similar approach may be extended to sPLA2. Woolford et al. [67] have conducted a systematic fragment screening against LpPLA2 in order to identify novel inhibitors. Among multiple fragment hits observed in different regions of the active site, there were some hits that bound in a pocket created by the movement of a protein side chain (reorientation of the Phe357 side chain) approximately 13 Å from the catalytic residue Ser273. The authors have optimized a fragment that bound in this pocket and generated a novel chemotype which does not interact with the catalytic residues but does inhibit catalytic activity. Mouchlis et al. [68], while investigating allosteric activation of LpPLA2 by membranes, have discovered that the hydrophobic pocket of the enzyme (used to bind the sn-2 acyl chain) has specific residues to bind oxidized fatty acids,—namely, through hydrogen bonding with His151, Tyr160 and His272, which are residues in the hydrophobic pocket. Thus, the presence of hydrogen bond donors at nine carbon atoms of the acyl chain of the substrate modulates binding and hydrolysis.

These findings, together with new data on PLA2-binding sites, could provide new small-molecule inhibitors which could be used to regulate enzyme activity, selectivity or both.

## 7. Conclusions

Secreted PLA2s are essential enzymes; however, their activity is extremely difficult to control. The involvement of sPLA2s in the development of various pathologies has led to a large-scale search for inhibitors of these enzymes. There are several possibilities to control PLA2 activity without acting on the catalytic site. They are: (1) preventing PLA2 from attaching to the membrane surface; (2) binding to an external protein which blocks hydrolytic activity; (3) preventing PLA2 from orienting properly on the membrane surface; and (4) preventing substrate binding to the enzyme, keeping the catalytic site unaltered.

Preventing PLA2 binding to the membrane surface is based on classical works on PLA2 kinetics originating from the 1990s. Today, there are several pieces of evidence suggesting that the approach is valid in vivo. Keeping in mind recent progress in LNP development, we believe that the approach is very promising. Binding PLA2 to an external protein which blocks hydrolytic activity is a topic under thorough investigation, which provides many details on PLA2 binding. We can summarize that the data on PLA2–protein interactions could serve at least as a starting point in the search for new molecules that interact with the PLA2 surface. Preventing PLA2 from orienting properly on the membrane surface is currently a concept which could be transformed into a promising approach to PLA2 activity regulation in the near future. Preventing substrate binding to the enzyme and keeping the catalytic site unaltered is based on docking and classic medicinal chemistry. Together with new data on PLA2 binding sites, it could provide new small molecules which could be used to regulate enzyme activity.

## Figures and Tables

**Figure 1 membranes-13-00618-f001:**
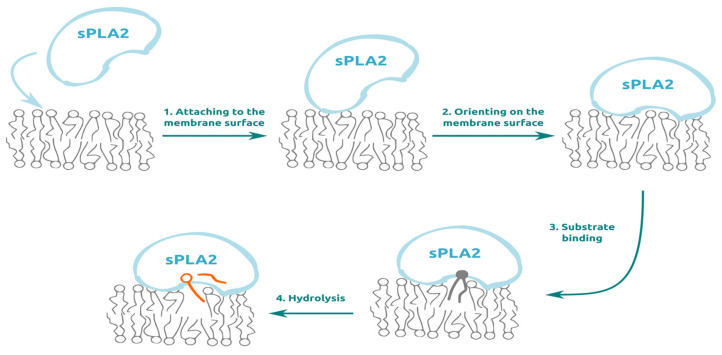
Mechanism of sPLA2 action. sPLA2 passes through multiple steps before it can hydrolyse the substrate. sPLA2 from the aqueous milieu attaches to the membrane surface (1). Once attached, PLA2 orients on the bilayer surface, thus making substrate binding possible (2). The properly oriented enzyme binds the substrate to the bilayer (3) and hydrolyses the substrate (4).

**Figure 2 membranes-13-00618-f002:**
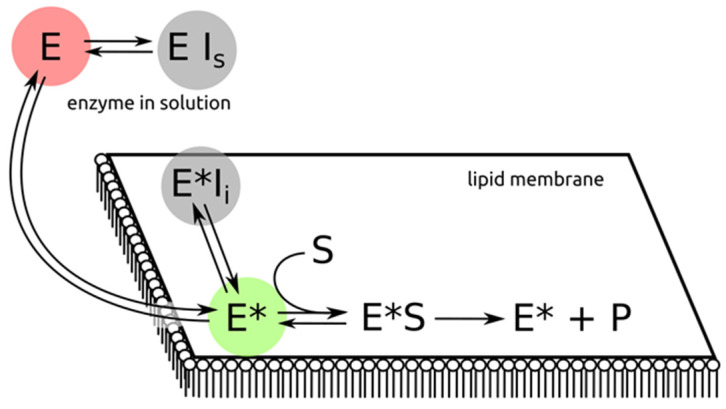
A kinetic scheme illustrating the key features of interfacial catalysis by sPLA2 and possible points for inhibition. E, enzyme in the solution; E*, enzyme bound to the bilayer; I_i_, inhibitor at the interface; I_s_, inhibitor in the solution; S, lipid substrate; P, products of the reaction.

**Figure 3 membranes-13-00618-f003:**
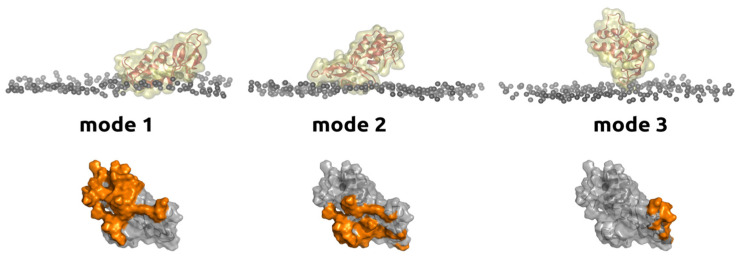
The target for inhibitors can be not only the catalytic site, but also other regions of the PLA2 protein. For example, even membrane binding can be carried out in three different ways, in each of which, theoretically, a protein can be fixed and taken out of its active state. Example of bvPLA2 on POPC membrane. Orange colour highlights membrane-binding regions of the enzyme in different modes.

**Figure 4 membranes-13-00618-f004:**
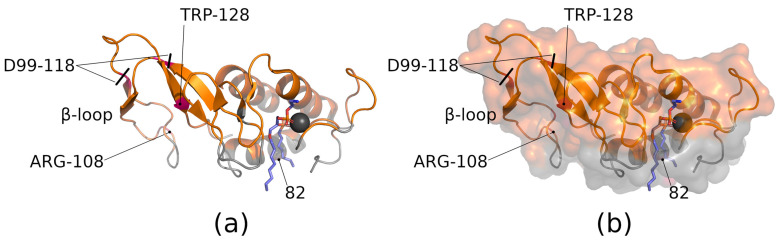
bvPLA2 with annotated parts and regions. (**a**) Ribbon view. (**b**) Surface view. Part of the enzyme contacting the bilayer (IBS site, mode 1) is coloured grey; Trp128 residue is outside IBS area. Substrate (blue) and calcium ion (black) mark the catalytic site.

## Data Availability

Not applicable.

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
