# Peer review of "Alternative Targets for sPLA2 Activity: Role of Membrane-Enzyme Interactions"

_membranes, 2023, doi:10.3390/membranes13070618_

Round 1

Reviewer 1 Report

This manuscript reviews alternative ways to suppress the function of secreted phospholipase A2 enzymes (sPLA2s) that do not involve direct binding to their catalytic site.The authors discuss a few possible targets including the binding of the enzyme to the membrane, its proper orientation once membrane-bound and indirectly interfering with the binding of the substrate. 

The review is well organized and the figures are clear and help the reader follow the discussion in the text. The manuscript needs language editing and there are some parts that need clarification as summarized below but overall the review is informative and should be of interest to the Membranes community.

Specific comments:

- I would suggest to reformat the title to more clearly and succinctly communicate the subject of the review. Something along the lines of “Alternative targets for sPLA2 activity: Role of membrane-enzyme interactions” or similar

- Please define sPLA2 in the abstract when first introduced

- I would suggest rewording “new sPLA2 inhibitors” on Line 18 to “new types of sPLA2 inhibitors”. Otherwise it sounds like more of the catalytic-site-targeting inhibitors. 

- Lines 42-44: failure in the clinical study may also be due to side effects of the drug so I would not limit the interpretation of the result to only the two mentioned cases.

- I would suggest rewording the sentence on Lines 112-113 to something like “Extensive work has been done on PLA2 kinetics with more than 200 publications on the topic from just three research groups [references].”

- In the discussion about neutral diluents, it does not become clear how much of the diluent needs to be added to the externally introduced liposome to make it “effectively unhydrolysable” (Lines 119-120). Can you please clarify?

- Line 124: Not clear how one can control the reaction by just monitoring the pH of the solution. Do you mean actively maintaining constant pH during the reaction? Please clarify.

- Lines 146-147: It is not clear how the experiment was conducted. “When using DTPC” — does this mean the enzyme did not bind to DTPC liposomes and instead hydrolyzed DMPM? It would help to briefly describe the experiment: what was added to what?

- In multiple places in the text there are references to particular parts/regions of the enzyme (e.g. beta-loop L185, Trp128 L268, sets of residues Lines 322 and 326). It would be helpful if the authors show a representation of the general structure of the enzyme and highlight these different regions so the reader can see where they are all located with respect to each other and to the catalytic site.

- Line 200: not clear why the binding affinity decreased. What is the experiment and what is being varied?

- Lines 203-206: the goal of this section 4 is to discuss the inhibitory binding of sPLA2 to other proteins as a potential way of suppressing its function. However, this whole paragraph seems to imply only that complexes of sPLA2 and other proteins have been shown to have undesirable negative effects. It is not clear what point is being made: that this is not a good approach or that it would be challenging since once proteins with inhibitory effect have been found, the role of the complex also needs to be considered? Please clarify. 

- Is the M-type sPLA2 receptor membrane-bound or soluble? It would help to clarify.

- Lines 237-240: This paragraph seems to describe the beneficial effects of PLA2 for preventing SARS-CoV-2 infection. The connection with this very relevant virus is interesting but this discussion seems misplaced here since the review is focused on ways to suppress the function of PLA2 given its many negative effects. If there are such positive effects then it may be best to mention them early in the beginning, then say however there are many negative consequences of PLA2 activity which require the development of inhibitors and thus this review etc. 

- Lines 245-247: The wording is confusing. For example, “If the IBS is involved in the process”: in what process? How can protein binding affect transfer of sPLA2 to endosomes? Can you give an example?

- In section 5, when discussing the different orientations of sPLA2 with respect to the membrane, can you comment on how much they depend on the lipid composition? Do the modes of binding of the enzyme to a bilayer change depending on what lipids are there and if so, how can this approach be generalized to work in cell membranes with more complex lipid compositions than the model membranes in which the discussed experiments take place?

- The idea about blocking substrate binding without interfering with the catalytic site is not very unintuitive. It would help if you briefly describe the main intent of this approach. Is it to design fragments that bind to the enzyme in regions around the catalytic site thus preventing the main lipid substrate from binding due to the lack of 'space' (like in the given example)? Are there any other ways in which this can be achieved?

The text is overall readable but there are multiple typos and grammatical errors that need to be fixed. Also, there are a few places where it is hard to understand the message being communicated.

Reviewer 2 Report

The article proposes new strategies to inhibit PLA2s independently of the catalytic site. Its idea is to prevent PLA2 interaction/orientation to the membrane and hydrolysis by interaction to different sites.

As a review, I found the text clear and fluid, with very good agreement between ideas. I think it will be useful for non-specialists and for newcomers.

English should be improved, I found minor issues, but I may have spotted all of them:
- the word "simmilar";
- Phrase in line 204-206 and page 6 seems to be missing a verb;
- Phrase in line 225 and page 6, "this proteins are not belong to the Ig family", change "are" for "do";
- Phrase in line 246 and page 7, change "than" for "then";
- bee venom PLA2 (bvPLA2) abbreviation is shown twice in text
- reference numbers duplicated in reference part.
